# Recovery of Ag, Au, and Pt from Printed Circuit Boards by Pressure Leaching

Guadalupe Martinez-Ballesteros [1], Jesus Leobardo Valenzuela-García [1,*], Agustin Gómez-Alvarez [1], Martin Antonio Encinas-Romero [1], Flerida Adriana Mejía-Zamudio [1], Aaron de Jesús Rosas-Durazo [2] and Roberto Valenzuela-Frisby [3]

[1] Department of Chemical Engineering and Metallurgy, University of Sonora, Hermosillo 83000, Mexico; guadalupe.martinezballesteros@gmail.com (G.M.-B.); agustin.gomez@unison.mx (A.G.-A.); martin.encinas@unison.mx (M.A.E.-R.); flerida.mejia@unison.mx (F.A.M.-Z.)

[2] Department Biomedical Engineering, Unidad Hermosillo, Sonora State University, Hermosillo 83100, Mexico; a.rosas.durazo@gmail.com

[3] Retroworks of México S.A. of C.V., Fronteras 84320, Mexico; rmababi@gmail.com

* Correspondence: jesusleobardo.valenzuela@unison.mx; Tel.: +52-662-847-6298

**Abstract:** Reclamation of printed circuit boards (PCBs) to recover metals is gaining growing attention due to minerals being non-renewable resources. Currently, metals extraction from PCBs through an efficient and green method is still under investigation. The present investigation concerns the recycling of printed circuit boards using hydrometallurgical processes. First, the basic metals (Cu, Ni, Zn and Fe) were separated using a sulfuric acid solution at moderate temperatures. The remaining solids were characterized by SEM-EDS, whereby a high content of precious metals (Au, Ag and Pt) was observed. In the second stage, solids were leached with a solution of HCl and NaClO in a 1-L titanium reactor with varied oxygen pressure (0.2, 0.34 and 0.55 MPa), temperature (40, 50 and 80 °C) and concentration of HCl (2 and 4 M), obtaining extractions above 95% at [HCl] = 4 M, P = 0.34 MPa and T = 40 °C. The extraction increased depending on the concentration of HCl. Eh–pH diagrams for Ag–Cl–$H_2O$, Au–Cl–$H_2O$ and Pt–Cl–$H_2O$ were constructed to know the possible species in the solution.

**Keywords:** pressure leaching; precious metals; printed circuit boards; recycling

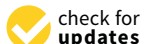



## 1. Introduction

Technology has recently advanced by leaps and bounds, and the production of electronic and electrical equipment (EEE) increases day by day. In a very short period, the most innovative of modern devices become obsolete at a faster rate than in the past. Therefore, the increase in waste originating from electrical and electronic equipment (WEEE) traces back to both private homes and professional users [1,2]. These materials are not properly treated when they reach the end of their useful life because they are sent untreated to landfills. Equipment components (heavy metals, plastics, polymers, ceramics, etc.) represent an important contamination source with severe ecosystem and health ramifications. Additionally, in recent years, concern has been growing for the environment worldwide [3]. Due to the increasing amounts of electronic waste generated in recent years, natural resources have depleted. Researchers have found it necessary to study various ways to recycle these wastes; as a result, their harmfulness has been reduced. Electronic waste is also a rich source of metals (basic, precious and rare earth) [3,4]. Printed circuit boards (PCBs) are the main components of electronic waste and are the primary source of metals of interest [5]. A PC's printed circuit board can contain up to 20% copper, 250 g/ton of gold and 110 g/ton of palladium [6]. These metals can be recovered via metallurgical processes, preferably via the hydrometallurgical route. With the pyrometallurgical route, very high temperatures are required to achieve incineration due to the properties of the base material of the PCBs (Bakelite, fiberglass resin, reinforced glass, ceramics or polymers). These

materials can withstand high temperatures. In addition, when they are incinerated, they generate dibenzo-p-dioxins and dibenzofurans (PCDD/Fs) and polybrominated diphenyl ethers (PBDD/Fs) [7–10].

The composition of WEEEs is much more complex than that of minerals, due to the physical and chemical characteristics of these materials; this permits the recovery of valuable metals being carried out in more stages for an efficient and environmentally sound processing of WEEE [11,12]. One way to recycle these is through hydrometallurgical processes since these do not generate as many gases and can separate metals selectively, due to the composition of waste PCBs, including plenty of toxic materials such as heavy metals, polyvinyl chloride (PVC) plastic and brominated flame retardants (BFR) [12]. Several researchers have studied the recovery of precious metals from WEEE. Gamez et al. studied the recovery of Au, Ag, Pd and Rb from discarded cell phone printed circuits. They used the cyanidation technique to extract Au, Ag and Pd by leaching with potassium hydroxide (KOH), nitric acid ($HNO_3$), sulfuric acid ($H_2SO_4$) and sodium chloride (NaCl). Recoveries of 95% Au, 82% Ag and 42% Pd were obtained [13]. Birloaga et al. carried out tests to recover Au by thiolation using 20 g/L $CS(NH_2)_2$, 6 g/L $Fe^{3+}$ and 10 g/L $H_2SO_4$, with stirring at 600 rpm; when performed at room temperature, 82% of the Au in the material dissolved [14]. Jadhav and Hocheng investigated the recovery of metals (Cu, Zn, Sn, Ni, Pb, Fe, Ag, Au and Pd) by leaching waste from PCBs via different leaching agents (HCl, $HNO_3$, $H_2SO_4$, $C_2H_4O_2$ and $C_6H_8O_7$). They designed a metal recovery process using one of these reagents, obtaining good results when HCl was used (taking 22 h to dissolve 100% of the studied metals) [15]. Torres and Lapidus studied the leaching of Au by thiourea at room temperature after pre-treatment for the elimination of Cu with inorganic acids (HCl, $HNO_3$ and $H_2SO_4$), extracting more than 90% Au [16]. Lei et al. studied the extraction of Pt and Au from anode-refining sludge derived from copper anodes via a roasting process followed by leaching with $H_2SO_4$ and sodium chloride (NaCl) as the chlorinating agent, recovering around 95% of Pt and more than 98% of Au [17].

HCl can be used as a leaching agent because it can dissolve precious metals, allowing for their recovery. Chlorination was widely applied in the 19th century, before the introduction of cyanidation, for the treatment of minerals containing fine gold and gold with sulfides, which could not be recovered by gravimetric concentrating. Chloride media has also been applied in electroplating processes since the early 19th century. Although chlorine–chloride media is no longer used to leach primary minerals, several processes have been proposed to treat refractory or semi-refractory minerals. As an alternative to cyanide, it has also been applied for the pre-treatment oxidation of some carbonaceous refractory minerals [18].

In the present investigation, the recovery of precious metals from PCBs was studied. Due to the complexity of the material and the presence of metallic alloys, hydrochloric acid was used as the leachate and sodium and oxygen hypochlorite as the oxidants. The thermodynamic characteristics of the leaching of metals and the effects of pressure and temperature on the extraction of Au, Ag and Pt are discussed using SEM-EDS and chemical analysis.

## 2. Materials and Methods

### 2.1. Materials

The materials were provided by Retroworks de México S.A. de C.V., located in Fronteras, Sonora, Mexico. The concentration of metals printed circuit boards (PCBs) was determined by the atomic absorption spectroscopy technique (Perkin Elmer model AAnalyst 400 atomic absorption equipment, PerkinElmer Inc, Waltham, MA, USA), and the metals concentration PCBs are shown in Table 1. In solutions, the pH and REDOX potential were analyzed using a Thermo Scientific Orion Star A111 pH benchtop meter (Thermo Fisher Scientific Inc, Waltham, MA, USA); this allowed us to construct a thermodynamic diagram of Eh–pH and determine the species in solution. The solid residues were analyzed to determine the characteristics of the material after leaching by scanning electron

microscopy and energy dispersive spectroscopy (SEM-EDS), using a Thermo Scientific Phenom Pro-X (Thermo Fisher Scientific Inc, Waltham, MA, USA). The Pourbaix Eh–pH diagrams were calculated using the HSC 6.0 program for each metal–electrolyte system under corresponding conditions. The chemicals (HCl$_{(l)}$ (37%), NaClO$_{(l)}$ (13%), NaCl$_{(s)}$, O$_{2(g)}$) were of analytical grade ($\geq$99.9%).

**Table 1.** Chemical composition of printed circuit boards.

| Metal | Ag | Au | Pt | Pd | Cu | Zn | Ni | Fe |
|---|---|---|---|---|---|---|---|---|
| Content | 170 (g/t) | 220 (g/t) | 2 (g/t) | 1.2 (g/t) | 13.14% | 0.02% | 4% | 4.9% |

*2.2. Methods*

The PCBs were subjected to a treatment with sodium hydroxide (10 M) to remove the epoxy material, for 24 h, and later, the PCBs were rinsed with water to remove the sodium hydroxide [15]. The epoxy material was removed so that it would not interfere with the leaching of metals. Then, they were cut into pieces of approximately 2 $\times$ 2 cm for further pulverization at a particle size of $-177$ $\mu$m. In the first stage, the dissolution of the basic metals was carried out by leaching with sulfuric acid (H$_2$SO$_4$) 2 M, pressure of 0.55 MPa and a temperature of 90 °C, obtaining recoveries greater than 91% Zn and 98% Cu and Ni. The chemical compositions (Table 2) of the solids after treatment with H$_2$SO$_4$ were determined by the atomic absorption spectroscopy technique (AAS).

**Table 2.** Chemical composition of printed circuit boards after pretreatment with H$_2$SO$_4$.

| Metal | Ag | Au | Pt | Pd | Cu | Zn | Ni | Fe |
|---|---|---|---|---|---|---|---|---|
| Content | 280 (g/t) | 320 (g/t) | 28 (g/t) | 12 (g/t) | 0.13% | 0.0016% | 0.004% | 0.43% |

In the second stage, the precious metal leaching was carried out with hydrochloric acid (HCl) as a leaching agent, using the oxidants 0.067 M sodium hypochlorite (NaClO) and 0.017 M (M = moles/L) sodium chloride (NaCl), present at 20% solids, as well as tap water. We added the pulp to the Titanium PARR Reactor (capacity 1 L) and processed it at 600 rpm for 2 h; varying the concentration of HCl (2 M and 4 M), the oxygen partial pressure (0.2, 0.34 and 0.55 MPa), and the temperature (40, 55, and 70 °C), the pressure inside the reactor remained constant due to the injection of oxygen and the temperature due to the heating jacket of the titanium PARR reactor. After the established contact time, the solution was filtered to separate the solid and the liquid. The solutions obtained were analyzed by the atomic absorption spectroscopy technique (AAS) to determine their metallic concentration, while solid residues were analyzed to determine the characteristics of the material by SEM-EDS. The flowsheet of metals selective leaching from PCBs is given in Figure 1.

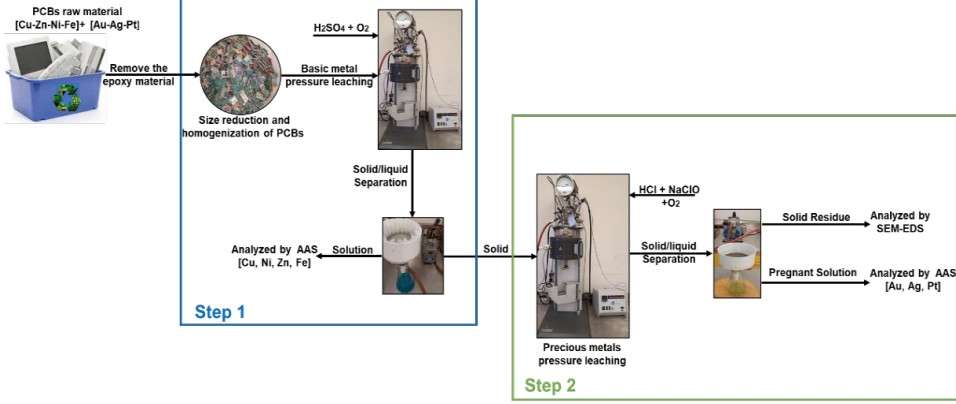

**Figure 1.** Flowsheet of metals selective leaching from PCBs.

## 3. Results

### *3.1. Pressure Leaching*

3.1.1. Silver

Figure 2 shows the effect on silver leaching when HCl concentration (2 and 4 M), pressure (0.2, 0.34, and 0.55 MPa) and temperature are varied (30, 55, and 70 °C). Figure 2A shows Ag extraction (%) as a function of pressure (MPa) at different temperatures and 2 M HCl. The best leaching conditions were reached when conditions of 40 °C and 0.34 MPa were used. The extraction of Ag under these conditions was above 50%. Figure 2B shows the graph of Ag extraction (%) as a function of pressure and temperature at 4 M HCl. An efficiency of Ag extraction greater than 98% was obtained when a temperature of 40 °C and a pressure of 0.34 MPa were used.

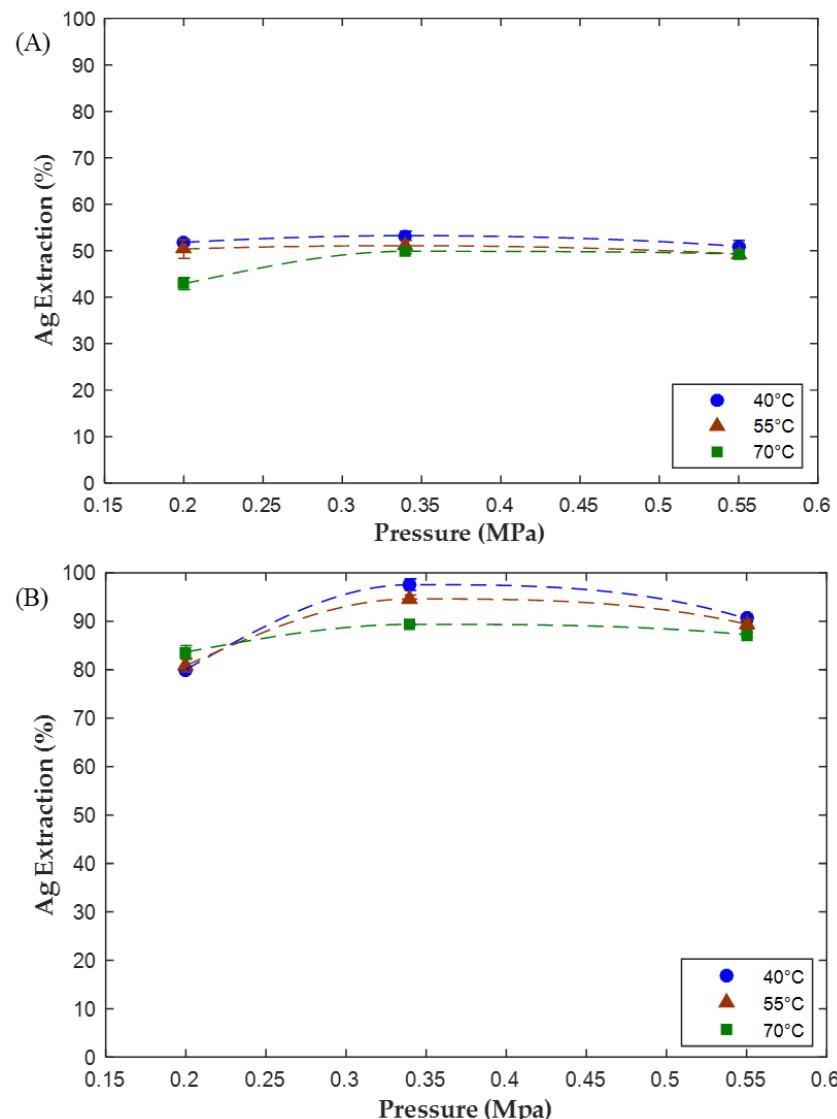

**Figure 2.** Ag extraction (%) as a function of pressure (MPa), at different temperatures, (**A**) at 2 M HCl and (**B**) 4 M HCl.

3.1.2. Gold

Figure 3 shows the gold leaching when the concentration of HCl (2 M and 4 M), the pressure (0.2, 0.34 and 0.55 MPa) and the temperature (40, 55, and 70 °C) were varied. Figure 3A shows Au extraction (%) as a function of pressure (MPa) at different temperatures and 2 M HCl. The extraction percentage is inversely proportional to the temperature. The

results show an Au extraction leaching efficiency of greater than 70% when a temperature of 40 °C and 0.34 MPa of pressure were used. Temperature affects the leaching of metals because HCl begins to decompose at temperatures higher than 53 °C [19]. On the other hand, at this concentration and 70 °C, not even 30% recovery is achieved for this metal. Figure 3B shows the graph of Au extraction (%) as a function of pressure and temperature at 4 M HCl. Au extraction was greater than 95% when a temperature of 40 °C and a pressure of 0.34 MPa were used.

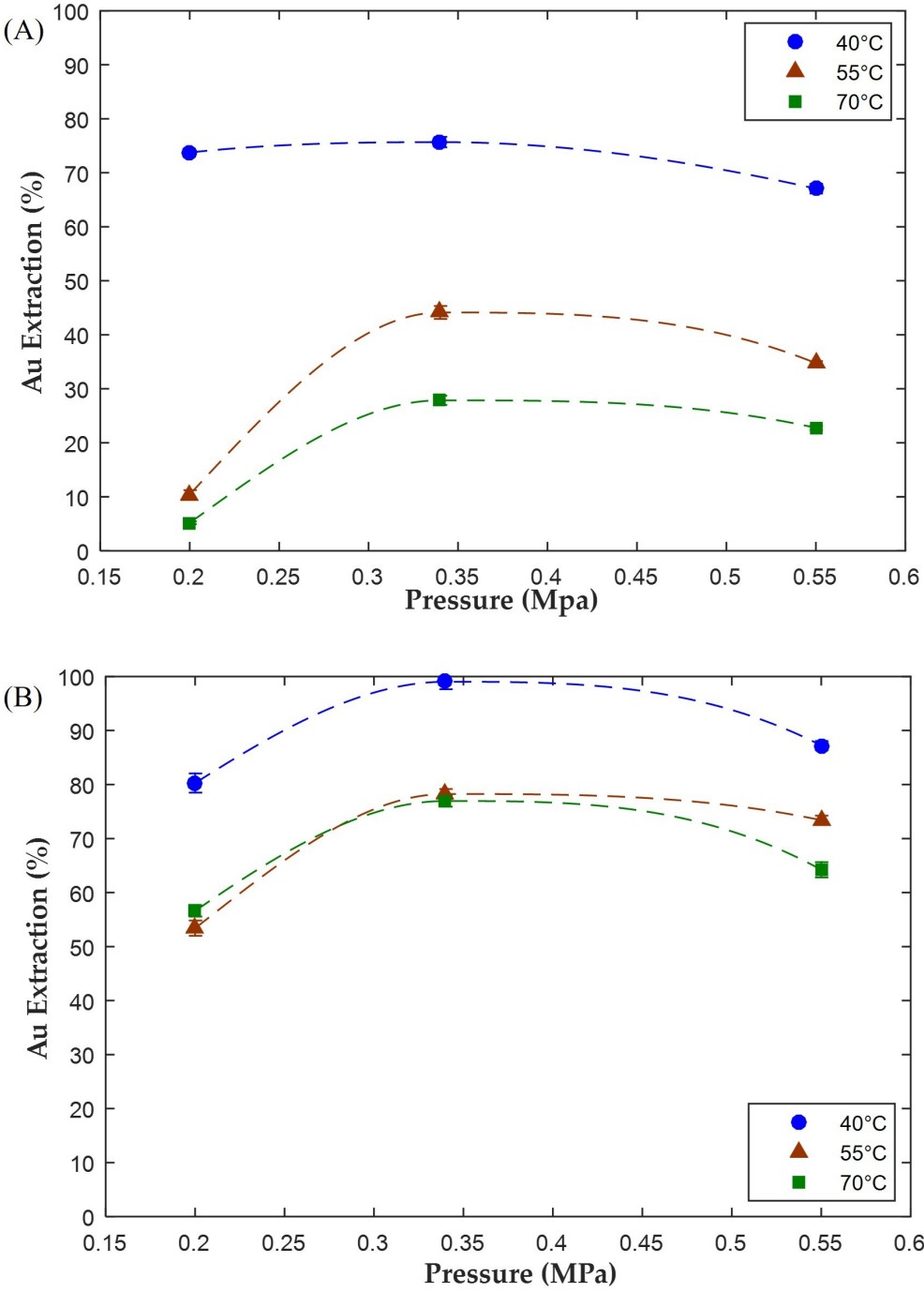

**Figure 3.** Au extraction (%) as a function of pressure (MPa), at different temperatures: (**A**) at 2 M HCl and (**B**) 4 M HCl.

The Solubility of HCl in $H_2O$ at higher temperatures, by increasing the concentration to about 0.1 M $Cl^-$, is not perceptible; the chloride activity becomes sufficient to stabilize

the $(AuCl_2)^-$ complex. Again, the acidity is shown to be important in maintaining the dissolved gold in the solution. As the solution temperature increases, the gold leaching efficiency also increases within limits. The effect of temperature on gold extraction from the gold sample dissolved in a solution containing 0.5% NaOCl, 5% HCl and 1 g/L NaCl was 82% of the gold was extracted in one hour at 27 °C, 95% at 40 °C and 90% at 55 °C. Further increasing the solution temperature to 70 °C or higher resulted in decreased gold extraction to 67%. This may be due to the greatly decreased chlorine dissolubility in water at a higher temperature, which agrees with the references [19,20].

### 3.1.3. Platinum

Figure 4 shows the effect on platinum leaching when the HCl concentration (2 M and 4 M), the pressure (0.2, 0.34 and 0.55 MPa) and the temperature (40, 55, and 70 °C) varied. Figure 4A shows Pt extraction (%) as a function of pressure (MPa) at different temperatures and 2 M HCl. Extraction is inversely proportional to temperature, whereby a Pt extraction above 65% was obtained when using a temperature of 55 °C and a pressure of 0.34 MPa. Temperature affects the leaching of metals because HCl begins to decompose at temperatures above 53 °C [19,20]. Figure 4B shows Pt extraction (%) as a function of pressure and temperature at 4 M HCl. A Pt extraction above 90% with a temperature of 40 °C and 0.34 MPa pressure was obtained. The leaching efficiency was greater at a concentration of 4 M HCl.

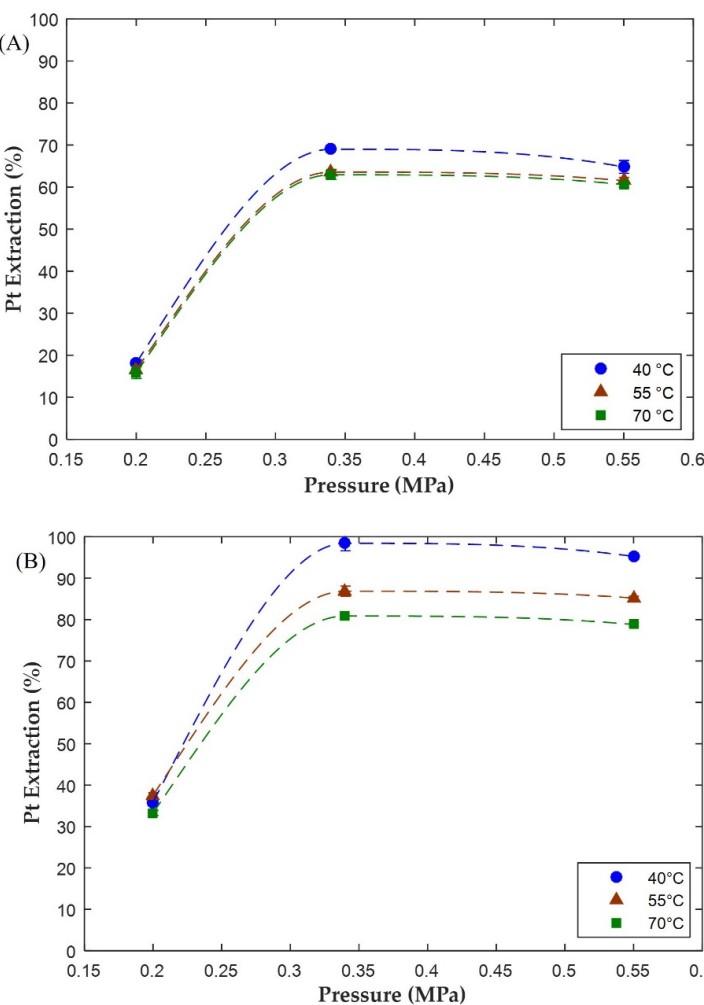

**Figure 4.** Pt extraction (%) as a function of pressure (MPa), at different temperatures: (**A**) at 2 M HCl and (**B**) 4 M HCl.

Table 3 shows the variation in the extraction (%) for the metals under study, Ag, Au and Pt, as a function of temperature, at a pressure of 0.34 MPa and concentration of 4 M HCl.

**Table 3.** Extraction (%) of Ag, Au and Pt = f (Temperature), at [HCl] = 4 M and P = 0.34 MPa.

| Time (min) | Ag Extraction (%) | | | Au Extraction (%) | | | Pt Extraction (%) | | |
|---|---|---|---|---|---|---|---|---|---|
| | 40 °C | 55 °C | 70 °C | 40 °C | 55 °C | 70 °C | 40 °C | 55 °C | 70 °C |
| 10 | 52.4 | 52.4 | 52.4 | 30.0 | 30.0 | 30.0 | 29.7 | 29.7 | 29.7 |
| 20 | 71.3 | 83.5 | 80.0 | 76.3 | 56.9 | 35.2 | 48.1 | 46.8 | 43.3 |
| 30 | 84.1 | 85.7 | 81.4 | 81.9 | 73.2 | 53.8 | 56.0 | 54.5 | 54.6 |
| 40 | 88.4 | 88.9 | 82.9 | 88.2 | 80.3 | 58.0 | 71.1 | 62.4 | 63.5 |
| 60 | 94.5 | 92.8 | 84.9 | 95.9 | 90.7 | 67.7 | 81.3 | 68.6 | 71.7 |
| 90 | 97.3 | 94.3 | 86.8 | 98.8 | 96.6 | 77.5 | 94.1 | 76.9 | 78.9 |
| 120 | 97.5 | 95.0 | 87.0 | 99.0 | 97.0 | 78.2 | 97.9 | 78.2 | 79.2 |

### 3.2. Thermodynamic Analysis

Precious metals (Ag, Au and Pt) have low chemical activity and are difficult to dissolve. The reactions that occur during the leaching of these metals or alloy with sodium chloride, hydrochloric acid, sodium hypochlorite and oxygen are indicated in Equations (1) to (6), considering that they are in metallic form on PCBs.

The leaching of gold occurs in two stages, as shown in Equations (1) and (2):

$$2Au^\circ_{(s)} + 4HCl_{(aq)} + ClO^-_{(aq)} + Cl^-_{(aq)} + \frac{1}{2}O_{2(g)} \rightarrow 2AuCl^-_{2\,(aq)} + Cl_{2(aq)} + 2H_2O \, \Delta G_{40°C} = 132.97 \frac{kJ}{mol} \tag{1}$$

$$2AuCl^-_{2(aq)} + Cl_{2(aq)} \rightarrow 2AuCl^-_{4(aq)} \, \Delta G_{40°C} = -162.61 \frac{kJ}{mol} \tag{2}$$

Silver leaching occurs in two stages, as shown in Equations (3) and (4):

$$2Ag^\circ_{(s)} + 4HCl_{(aq)} + ClO^-_{(aq)} + Cl^-_{(aq)} + \frac{1}{2}O_{2(g)} \rightarrow 2AgCl^-_{2(aq)} + Cl_{2(aq)} + 2H_2O \, \Delta G_{40°C} = 14.46 \frac{kJ}{mol} \tag{3}$$

$$2AgCl^-_{2(aq)} + Cl_{2(aq)} \rightarrow 2AgCl^{2-}_{3(aq)} \, \Delta G_{40°C} = -256.36 \frac{kJ}{mol} \tag{4}$$

To carry out the dissolution of platinum, the following reactions occur in (5) and (6):

$$Pt^\circ_{(s)} + 4HCl_{(aq)} + ClO^-_{(aq)} + Cl^-_{(aq)} + 1/2O_{2(g)} \rightarrow PtCl^{2-}_{4(aq)} + Cl_{2(aq)} + 2H_2O \, \Delta G_{40°C} = 86.61 \frac{kJ}{mol} \tag{5}$$

$$PtCl^{2-}_{4(aq)} + Cl_{2(aq)} \rightarrow PtCl^{2-}_{6(aq)} \, \Delta G_{40°C} = -119.39 \frac{kJ}{mol} \tag{6}$$

Figures 5–7 show Eh–pH Pourbaix diagrams, which were constructed for each metal–electrolyte system using HSC 6.0 software with the conditions of 40 °C, 0.34 MPa and the corresponding metal concentration. The two indicated points represent the species present at the different molarities of HCl used in the experimental tests, according to REDOX potential (Eh) and pH (2 M (Eh = 0.8 v, pH = −1) and 4 M (Eh = 1.3 v, pH = −1.7)). Figure 5 shows the species diagram for the Ag–Cl–H$_2$O system, wherein the location of the points suggests the existing species in the solution is aqueous silver chloride (AgCl$_3$$^{2-}$). With HCl at 4 M, there is a greater area of stability, and the leaching efficiency is above 97%.

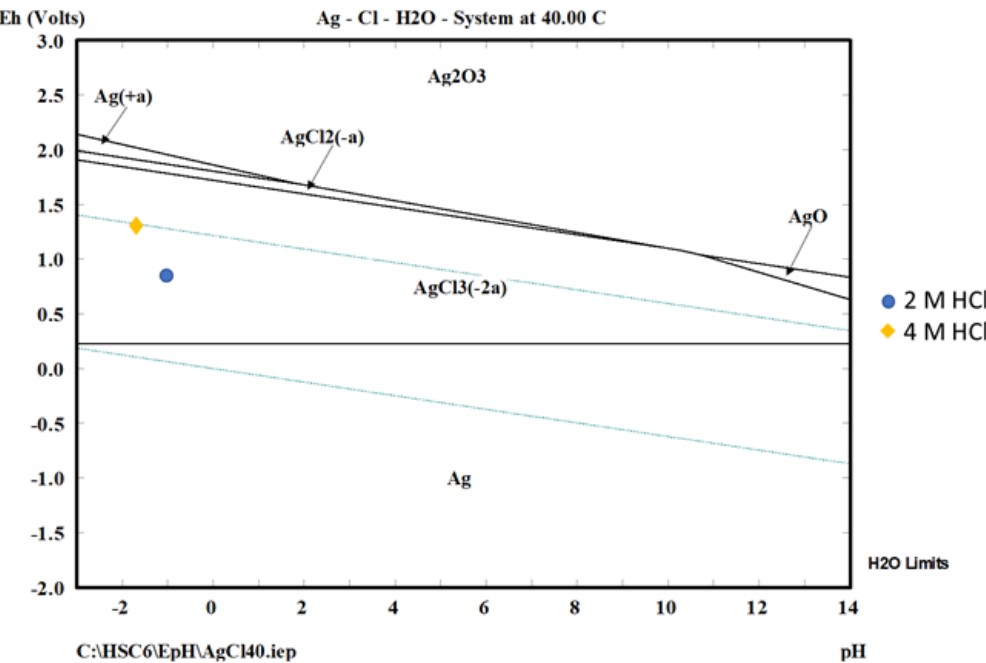

**Figure 5.** Eh–pH diagram for the Ag–Cl–H$_2$O system at [Ag] = 0.00048 M, 40 °C and 0.34 MPa.

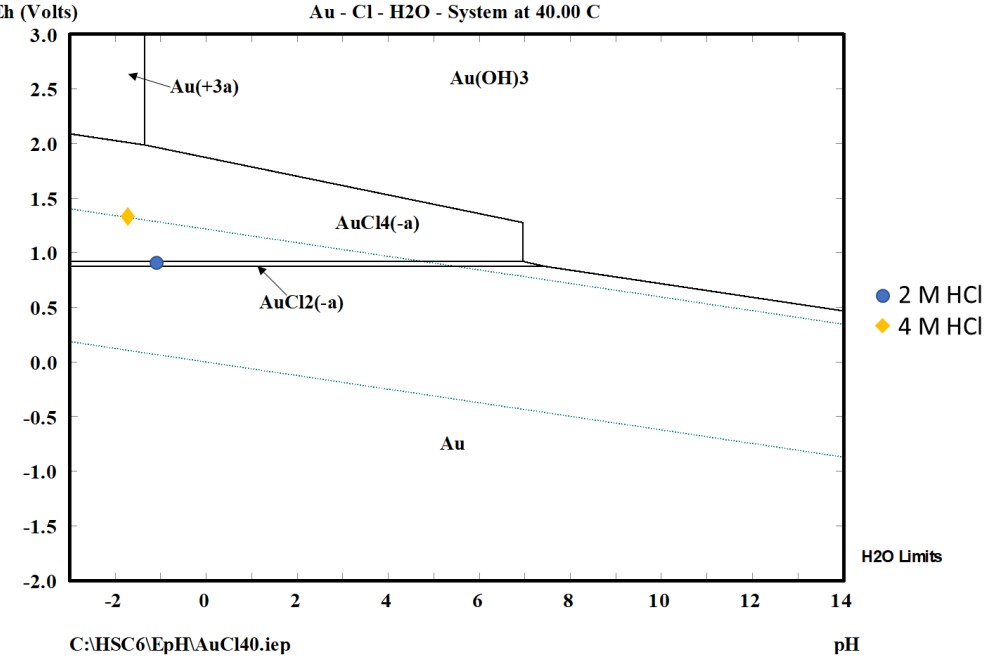

**Figure 6.** Eh–pH diagram for the Au–Cl–H$_2$O system, with [Au] = 0.00036 M, 40 °C and 0.34 MPa.

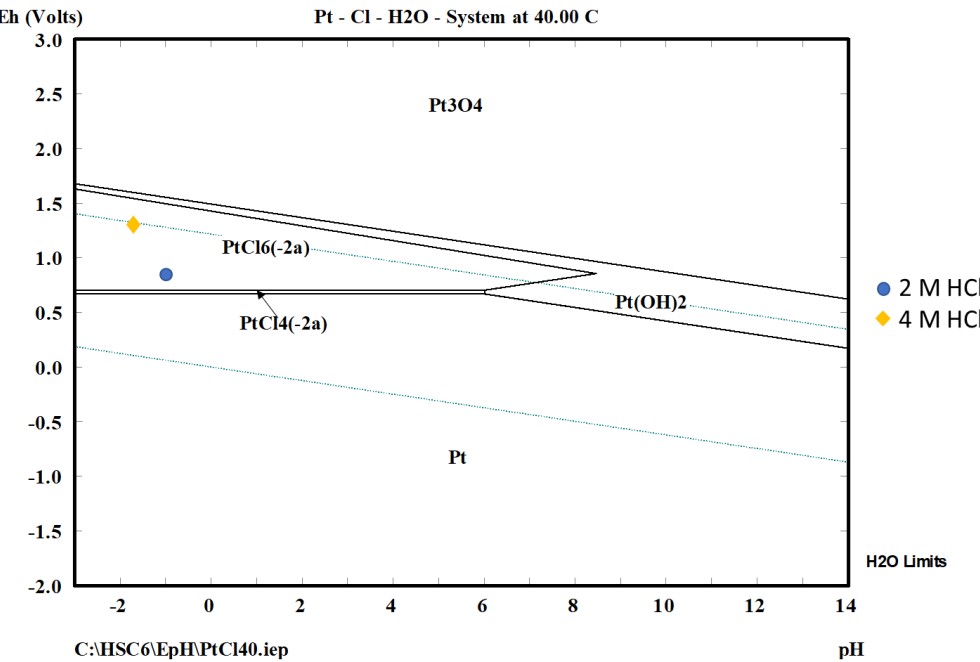

**Figure 7.** Eh–pH diagram for the Pt–Cl–H$_2$O system with [Pt] = 0.000025 M, 40 °C and 0.34 MPa.

Figure 6 shows the species diagram for the Au–Cl–H$_2$O system, wherein the locations of the points suggest the existing species in the solution is aqueous gold chloride (AuCl$_4^-$). However, with HCl at 4 M, based on the greater species stability, the extraction efficiency is above 95%.

Figure 7 shows the species diagram for the Pt–Cl–H$_2$O system, wherein the locations of the points suggest the existing species in the solution is the aqueous chloride–platinum complex. With a pH lower than 7 and a redox potential above 0.7, the soluble complex (PtCl$_6^{2-}$) is stable. However, with HCl at 4 M, the area of stability is greater since the leaching efficiency of platinum in these conditions was greater than 98%.

*3.3. Study of PCBs Material from the Leaching Tests by SEM-EDS*

Figure 8 shows the SEM micrograph and microanalysis of the PCBs material after pre-treatment with H$_2$SO$_4$ and before leaching with HCl for the recovery of Ag, Au and Pt. Results show that the morphology of the material is very diverse and contains elements such as Au, Pd, W, Ti, Sn and Br. It should be noted that the presence of other elements is due to the PCBs base composition; for example, Br presence is caused by the Brominated flame retardants (BFRs) used in the plastic housings of electronic equipment and in PCBs to prevent flammability [21].

Figure 9 shows the SEM micrograph and microanalysis of the sample leached with 4 M HCl at 0.34 MPa and 40 °C, from which the highest percentage of precious metals was extracted. Additionally, the non-evidence of the metals Ag, Au and Pt were confirmed, since they were leached by means of HCl, NaClO and O$_2$. The morphology of the material is very diverse, and the identified elements were Al, Br, Ti, Mg, Si and Cl; the latter is present in the reagents used for leaching. It should be noted that the presence of the other elements is due to the PCBs base composition as mentioned earlier, for example, Ba, which is used to protect users from radiation.

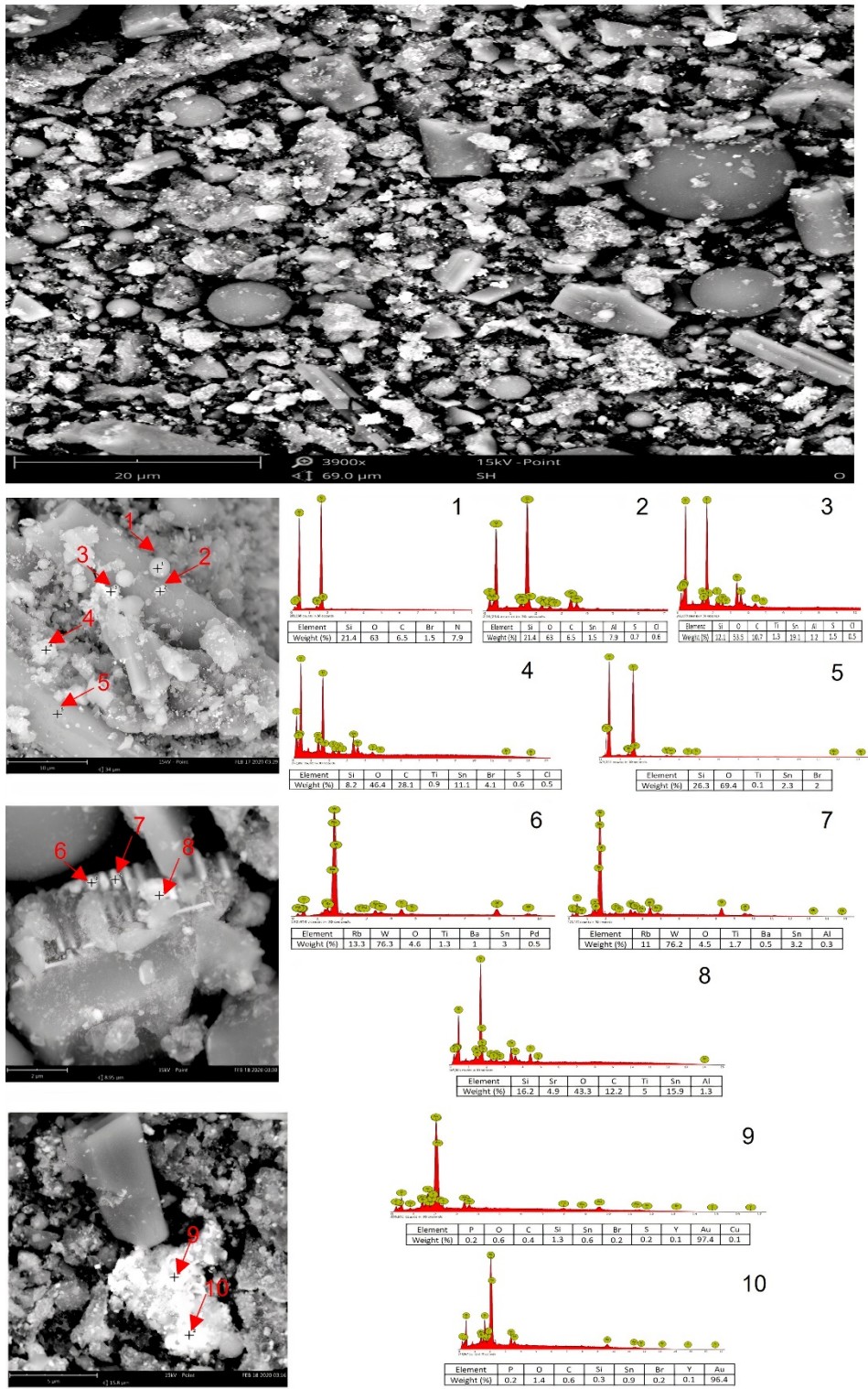

**Figure 8.** Scanning electron microscopy (SEM-EDS) images, micrograph and microanalysis of the PCBs sample before HCl leaching. The numbers (1–10) mean punctual microanalysis.

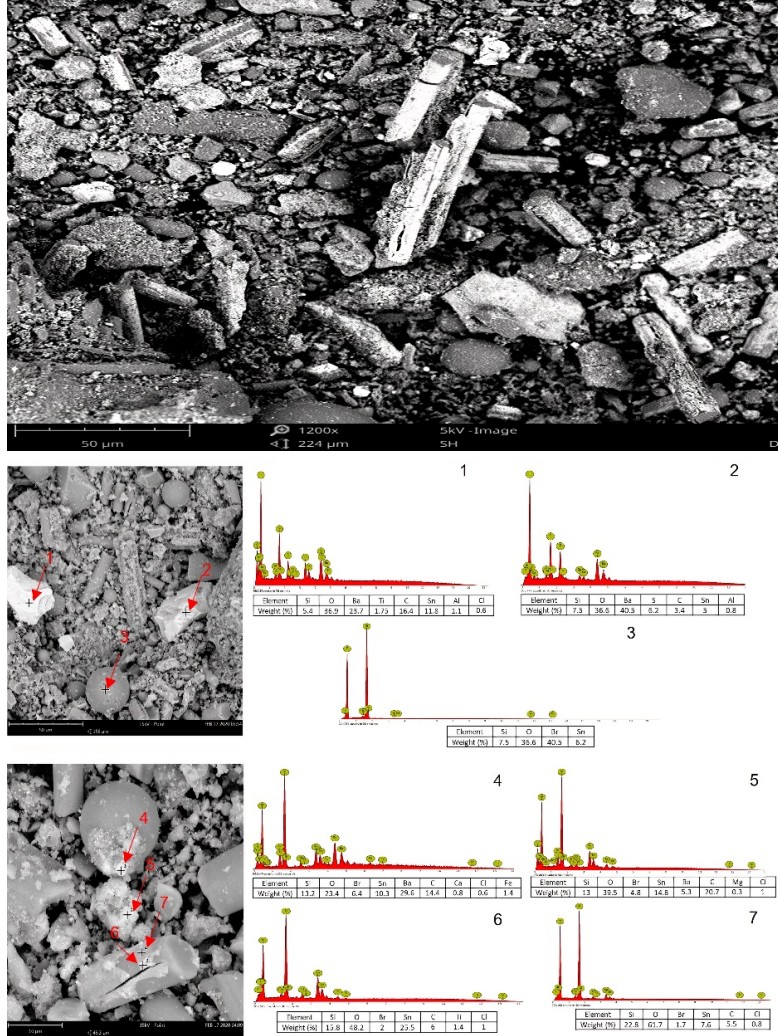

**Figure 9.** Scanning electron microscopy (SEM-EDS) images, micrograph and microanalysis of the PCBs sample leached with 4 M HCl at 0.34 MPa and 40 °C. The numbers (1–7) mean punctual microanalysis.

## 4. Discussion

Gold and platinum, as precious metals, have a low chemical activity, showing a high resistance to dissolution. A chlorination leaching method is directly used to extract Pt, Ag and Au from waste PCBs, in which the Pt and Au are chlorinated and transferred to acid soluble $PtCl_6^{2-}$, $AgCl_3^{2-}$ and $AuCl_4^-$ through Equations (1)–(6) in a solution of NaCl–HCl–NaClO. The stability Eh–pH diagrams for systems of Ag–Cl–$H_2O$, Au–Cl–$H_2O$ and Pt–Cl–$H_2O$ at 40 °C are presented in Figures 5–7, respectively. The potential for Ag leachable varies from 0.2 V to 1.9 V over a wide pH range, Au leachable ranges from 0.96 V to 2.12 V with the pH from −2 to 7.05, as seen in Figure 6, and that for Pt extends from 0.65 V to 1.65 V with the pH from −2 to 7.7, as shown in Figure 7. To achieve these high potentials for obtaining high leaching efficiency of Au and Pt, much NaClO should be employed in the leaching process. However, it causes the utilization efficiency of chlorine to be decreased due to the higher emission of $Cl_2$ during the leaching process.

SEM micrographs of the raw material before leaching with hydrochloric acid show that the morphology is very diverse. The microanalysis shows that it contains several chemical elements: Au, Pd, Rb, W, Ti, Sn and Br, of which Au and Pd were of interest in this research, and with the microanalysis, it is shown that there are particles with these metals of interest. Likewise, in the micrograph of the leached solid where a higher percentage of extraction of

the metals of interest was obtained, it is observed that the morphology of this material is very diverse, and the particles analyzed by microanalysis contain Ba, Br, Cl, O, Ti, Sn and Si, of which none were an item of interest.

In both micrographs of the material, before and after the leaching of precious metals with hydrochloric acid, spherical and cylindrical particles can be observed. By microanalysis with EDS, it was determined that its elemental composition is Si, O, C, Sn, Br and N; these elements form the basis of printed circuit boards. In the leaching of PCBs with the help of chlorine gas in aqueous medium, a high recovery of gold could be obtained, but chlorine gas causes serious corrosion problems of the equipment. [17]. Based on the results of this research, it is possible to take advantage of strongly oxidizing hypochlorite species, which can be generated by adding inorganic hypochlorite salts to a solution [18]. A chlorination leaching process using a NaCl–HCl–NaClO mixture has greater advantages, including faster gold leaching efficiency, reaching up to 95% in 1 h and 99% in 2 h. However, the leaching efficiency of platinum is low under optimal conditions favoring the leaching of Au, which causes a considerable amount of platinum to be distributed in the leaching residue and, consequently, the leaching efficiency of platinum can be difficult to increase. The temperature has an adverse effect on platinum extraction because with increasing temperature, platinum extraction decreases, which is in accordance with the references [19,20]. At a temperature of 70 °C, less metal extraction was obtained compared to lower temperatures, probably because of the significant reduction in the dissolubility of chlorine in water at high temperatures. The extraction of Au and Ag decreased with the increase in pressure to 0.55 MPa, explained by the excessive amount of $H_2O$ generated. Therefore, the chlorine gas formation was reduced due to the transition of hydrochloric acid and chlorine, resulting in a decrease in metal extraction.

## 5. Conclusions

The results of this research revealed that more than 97% Ag, 99% Au and 98% Pt can be leached from the PCBs by an NaCl–HCl–NaClO mixed solution in 2 h, at pressure of 0.34 MPa, a temperature of 40 °C, HCl at 4 M and NaClO at 0.067 M.

The concentration of HCl and the temperature significantly affect the leaching of Au and Pt, as better results are obtained at a concentration of 4 M and a temperature of 40 °C. Based on the Pourbaix diagram of the Pt–Cl–$H_2O$ system, at 2 M HCl, Pt dissolves less effectively and, in the Pourbaix diagram of the Ag–Cl–$H_2O$ system, if this metal covers Ag or Au (or some other metal), it is not able to leach. Instead, at a HCl concentration of 4 M, this metal remains as an aqueous chloride and the percentage of extraction of Ag is higher.

Under the experimental conditions, the Eh–pH diagrams confirmed extraction of these metals, mainly because they ground thermodynamically favorable conditions for metal leaching.

According to the results of the laboratory tests, this research shows that it is feasible to recover precious metals from PCBs by leaching with hydrochloric acid and sodium hypochlorite at moderate temperatures and pressures. Recovery of major beneficial metals, Ag, Au and Pt, followed by production of minimal waste materials with known character, represent a great advantage for presenting a technological route, which is of the highest importance for the lower operating capacities. Further analysis of the process economy could give a more comprehensive overview if such a process represents a promising alternative for the present status dominated by the pyrometallurgical sector.

**Author Contributions:** Conceptualization, G.M.-B. and J.L.V.-G.; formal analysis, G.M.-B. and F.A.M.-Z.; investigation, G.M.-B., J.L.V.-G., and A.G.-A.; methodology, G.M.-B., J.L.V.-G., A.G.-A., and A.d.J.R.-D.; project administration, J.L.V.-G. and R.V.-F.; resources, M.A.E.-R., R.V.-F., and J.L.V.-G.; data curation, A.G.-A. and F.A.M.-Z.; formal analysis, G.M.-B. and J.L.V.-G.; supervision, J.L.V.-G. and A.G.-A.; validation, G.M.-B. and M.A.E.-R.; writing—original draft, G.M.-B.; writing—review and editing, J.L.V.-G., A.G.-A., A.d.J.R.-D., and F.A.M.-Z. All authors have read and agreed to the published version of the manuscript.

**Funding:** This research was funded by LANGEM, Grant Number 294889, National Laboratory of Geochemistry and Mineralogy in México and CONACYT (National Council of Science and Technology) for graduate scholarship of one author (G.M.-B.).



**Institutional Review Board Statement:** Not applicable.

**Informed Consent Statement:** Not applicable.

**Data Availability Statement:** Not applicable.

**Acknowledgments:** Thanks to the Engineering Division, Chemical Engineering and Metallurgy Department, and the Geology Department of the University of Sonora, and Retroworks de México for their support in this research.

**Conflicts of Interest:** The authors declare no conflict of interest. The funders had no role in the design of the study; in the collection, analyses, or interpretation of data; in the writing of the manuscript; or in the decision to publish the results.

**Sample Availability:** Not available.

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
