# Peer review of "Recovery of Ag, Au, and Pt from Printed Circuit Boards by Pressure Leaching"

_recycling, doi:10.3390/recycling6040067_

Round 1

Reviewer 1 Report

The study concerns the leaching behaviors of Ag, Au, and Pt with HCl from used PBC and has found some interesting leaching behaviors. However, the study lacks in careful description of the system and narrow interpretation of the results without other possibilities which might be taking place in the system. The leaching behaviors of each element in the absence of other major metals present in the system under the similar conditions should have been referred to in order to explain the phenomena occurring in the system.

The authors need to explain the following aspects listed below:

The authors should mention how the pressure is defined: is it psia or psig? Furthermore, how was the pressure adjusted: was it done by adding air, N2, etc?

The authors should be aware that the solubility of HCl in water decreases with temperature and should take this into account in the discussion of the results. Furthermore, the authors should have explained how the consumption of HCl by other metals, presumably the major constituents in terms of the mass, present in the system might have an effect on the leaching of the metals concerned in this study.

The authors have chosen the leaching time at 2 hours and didn’t study the effect of time. It would be interesting to see if the 2 hour-leaching represents the equilibrium of the reaction or it represents a transient behavior. The authors have treated as if the time represents equilibrium since Eh-pH diagram does represent the equilibrium situation. Please explain.

With regard to Table 1: This table represents the composition of the PCB tested in this study, but it is very unfortunate that the table is far away from the true composition. What are the major constituents of the board which may consume HCl and its effect on the overall behavior of leaching process. For example, the major constituent such as copper may consume most of HCl especially at high temperature, which may cause a deficiency of HCl for reaction of the metals concerned in this study. Please explain.

Reviewer 2 Report

Recovery Ag, Au, and Pt from Printed Circuit Boards by Pressure Leaching

comments of the reviewer

  1. a P = 50 psi, 20
  2. concentration of HCl (2 M and 4 M), pressure (30, 50, 80 psi), and temperature (40, 55, 70 )97

the pressure unit should be in the SI system

1(pounds-force per square inch)= 6895Pa

145Psi=1MPa

  1. cause they are sent to landfills or landfills without any treatment. The type of equipment 32

what is the difference between a landfill and a landfill, can it be sent for recycling or to a landfill ??? /

  1. The material was provided by the company Retroworks de México S.A. de C.V., lo- 86 cated in Fronteras, Sonora, Mexico. The PCBs were subjected to a treatment with sodium 87 hydroxide (10 M) to remove the epoxy material [14]. Later they were cut into pieces of 88 approximately 2 x 2 cm for further pulverization. In the first stage, the dissolution of the 89 basic metals was carried out by leaching with sulfuric acid (H2SO4), pressure, and temper- 90 ature. T

NaOH is very corrosive and will not react with the metals in PCBs and leach them out. Were the PCBs washed with water after alkaline treatment? After all, PCB is treated with acid. Very weak, unclear description of the procedure for handling PCBs.

  1. Figure 2. Effect of pressure (psi) and temperature (°C) on Ag extraction (%) in the leaching of PCB, 132 A) at 2 M HCl and B) at 4 M HCl.

Rather, it would be more beneficial for the presentation of the results to present the relationship more clearly, e.g.% Au = f (Temp, and pressure - as legend) or% Au = f (pressure and Temp, - as legend). I would recommend moving away from 3-D in the absence of results in the table.

  1. Figure 8 shows the SEM micrograph and microanalysis of the PCBs material after 221

maybe an analysis in a few points or a map analysis, because what is in other elements / points of the sample (chemical composition?) whether the analyzed point is marked with an an. It is best to mineralize the sample and ICP analysis and then compare the PCB residue before and after the pressure dissolution.

  1. several metals: Au, Pd, Rb, W, Ti, Sn and Br. Likewise, it is appreciated in the micrograph 256

 tained; in which the morphology is very diverse and contains Br, Sn and Si, these metals 258

Bromine is a non-metal from the group of halogens in the periodic table.  silicon is a semi-metal

  1. The discussion of the results is very poor, the conclusions say little about the results

Round 2

Reviewer 1 Report

The paper describes some interesting results on the leaching behaviors of the three metals with the HCl system at different temperatures. However, the discussion of the phenomena taking place is vague and unclear. In particular, the effect of pressure on the leaching behavior is not clear at all.

First, what are the final concentrations of HCl at various temperatures? After all these concentrations and not the initial concentrations should be considered for the equilibrium phenomena such as Eh-pH diagrams and equilibrium concentration of each metal dissolved etc.

Second, the authors’ claim that the two hours of the leaching as the equilibrium time is not satisfactory. For example, the leaching of gold can go further should there be more time since the leaching has not reached a plateau as silver has.

The authors’ explanation of the pressure effect on the leaching efficiency is not at all clear. What do they mean by the effect of the pressure, is it a partial pressure of O2, HCl etc? or the total pressure, for example on the Eh-pH diagrams?  

To consider this in a more systematic way, let us consider Eqs 1 and 2 with the effect of the total pressure of the system. When a temperature was established, the authors suggested that they adjust the pressure by adding O2 to the system to get a pressure point and then add more O2 to another. By doing this, they not only change the total pressure of the system but also the partial pressure of O2 and HCl, and all other components in the system. Remembering the gas phase of the system may contain, HCl, O2, H2O, N2 etc, by changing the total pressure, they also change the partial pressure of each component. How does this effect the leaching of gold given by Eqs 1 and 2? Could the authors quantitively explain what the change of the partial pressure of each of the gases at a given temperature have an effect on the rate of dissolution of gold? More quantitative analysis would help understand what is happening in the system.

Reviewer 2 Report

Thank you for considering my comments. Now, in my opinion, paper meets the requirements for printing. 
